

# Towards efficient verifiable multi-keyword search over encrypted data based on blockchain

Wanshan Xu[1,2], Jianbiao Zhang[1,2], Yilin Yuan[1,2], Xiao Wang[3], Yanhui Liu[1,2] and Muhammad Irfan Khalid[1,2]

[1] Faculty of Information Technology, Beijing University of Technology, Beijing, China
[2] Beijing Key Laboratory of Trusted Computing, Beijing, China
[3] Department of Information Science and Technology, Tianjin University of Finance and Economics, Tianjin, China

## ABSTRACT

Searchable symmetric encryption (SSE) provides an effective way to search encrypted data stored on untrusted servers. When the server is not trusted, it is indispensable to verify the results returned by it. However, the existing SSE schemes either lack fairness in the verification of search results, or do not support the verification of multiple keywords. To address this, we designed a multi-keyword verifiable searchable symmetric encryption scheme based on blockchain, which provides an efficient multi-keyword search and fair verification of search results. We utilized bitmap to build a search index in order to improve search efficiency, and used blockchain to ensure fair verification of search results. The bitmap and hash function are combined to realize lightweight multi-keyword search result verification, compared with the existing verification schemes using public key cryptography primitives, our scheme reduces the verification time and improves the verification efficiency. In addition, our scheme supports the dynamic update of files and realizes the forward security in update. Finally, formal security analysis proves that our scheme is secure against Chosen-Keyword Attacks (CKA), experimental analysis demonstrations that our scheme is efficient and viable in practice.

Corresponding author
Jianbiao Zhang, zjb@bjut.edu.cn

## INTRODUCTION

With the development of artificial intelligence, the Internet of Things, the Internet of Vehicles and other emerging technologies, more and more enterprises and individuals outsource local data to the cloud, thereby reducing storage and management overhead. However, security and privacy concerns still hinder the deployment of the cloud storage system. Although data encryption can eradicate such concerns to some extent, it becomes difficult for users to search over the data.

Searchable symmetric encryption (SSE) provides an efficient mechanism to solve this, which enables users to search encrypted data efficiently without decryption. Since SSE was first proposed by *Song, Wagner & Perrig (2000)*, how to perform efficient and versatile

search on encrypted data has always been an important research direction. The existing SSE schemes mainly use linked lists and vectors to build indexes, the cloud server needs to traverse the whole list or vector to search for matching results during a query, which incurs high search overhead. In addition to efficient searching, dynamic updates are also very important in SSE. *Zhang, Katz & Papamanthou (2016)* has shown that adversaries can infer the critical information through the file injection attacks during the dynamic update of the SSE, while the forward-secure SSE can avoid this. Therefore, the forward security of the scheme must be fully considered when designing the SSE scheme.

Verifiability of the search results is another important research issue for SSE. Since the cloud server is untrusted, which may returns incorrect or incomplete results due to system failures or cost savings, so, it is necessary to verify the search results. In 2012, *Chai & Gong (2012)* proposed the concept of verifiable SSE (VSSE) and constructed a verifiable SSE scheme based on word tree. Following this work, a great many VSSE schemes are proposed *Kurosawa & Ohtaki (2012)*, *Sun et al. (2015)*, *Zhu, Liu & Wang (2016)*, *Liu et al. (2017)*, *Zhang et al. (2019)* and *Chen et al., 2021*). In these schemes, the verification is mainly performed by users, but the user may forge verification results to save costs, so the reliability of the verification cannot be guaranteed. To address this, some researchers (*Cai et al., 2018*; *Hu et al., 2018*; *Li et al., 2019*; *Guo, Zhang & Jia, 2020*) introduce blockchain into SSE to verify search results, which guarantees the fairness and reliability of the verification. Although blockchain achieves fair verification of search results, but the existing schemes are only for a single keyword, and there is little research on fair verification for multi-keywords.

In this paper, we introduce a verifiable multi-keyword SSE scheme based on blockchain, which can perform efficient multi-keyword search, ensures the fairness of verification, and supports the dynamic update of files. To our knowledge, this is the first scheme to verify the search results of multi-keywords fairly. In general, the contributions of this paper are summarized as follows:

- Our scheme realizes efficient multi-keyword search and verification of search results, at the same time, our scheme supports dynamic update of files and achieves forward security.
- Our scheme utilizes blockchain to verify the search results, ensuring the reliability and fairness of the verification results. Combining bitmap index and hash function, we realize lightweight multi-keyword verification to improve verification efficiency.
- We formally prove that our scheme is adaptively secure against CKA, and we conduct a series of experiments to evaluate the performance of our scheme

## RELATED WORKS

### Searchable symmetric encryption

Since SSE was proposed, a number of works have been done to improve search efficiency, rich expression and advanced security. The first SSE scheme (*Song, Wagner & Perrig, 2000*) enables users to search keywords through full-text scanning, search time increases linearly with the size of files, which is impractical and inefficient. To improve efficient, *Curtmola et al. (2006)* proposed an inverted index SSE, which achieves sub-linear search time, and

gives a definition of SSE security, but this scheme does not support dynamic operations. *Wang et al. (2010)* expanded the scheme of *Curtmola et al. (2006)* to support dynamic operations, and proved that the scheme was adaptively secure against chosen-keyword attacks (CKA2-secure). For the schemes that support dynamic operation, forward security is critically crucial. The research of *Cash et al. (2013)*, *Cash et al. (2015)* and *Zhang, Katz & Papamanthou (2016)* indicated that in the SSE scheme without forward security, the adversary can recover most of the sensitive information in ciphertext at a small cost, their research shows the importance of forward security.

Multi-keyword search is a crucial means to improve search efficiency. In single-keyword search scheme (*Song, Wagner & Perrig, 2000*; *Curtmola et al., 2006*; *Wang et al., 2010*; *Kamara, Papamanthou & Roeder, 2012*), the server returns some irrelevant results, while the multi-keyword search (*Cash et al., 2013*; *Lai et al., 2018*; *Xu et al., 2018*; *Wang et al., 2018*; *Liang et al., 2020*; *Hozhabr, Asghari & Javadi, 2021*; *Liang et al., 2021*) gains higher search accuracy and more accurate results. To further improve search efficiency, *Abdelraheem et al. (2016)* proposed an SSE scheme on encrypted bitmap indexes to support multi-keyword search, but requires two rounds of interactions with the cloud server. *Zuo et al. (2019)* proposed a secure SSE scheme based on bitmap index which supports dynamic operations with forward and backward security, but this scheme lacks the verification of the results.

## Verifiable searchable symmetric encryption

In SSE, it is necessary to verify the results since the server is untrusted. *Chai & Gong (2012)* proposed the concept of verifiable searchable symmetric encryption (VSSE) and constructed a VSSE scheme based on word tree. Along this direction, some other VSSE schemes (*Kurosawa & Ohtaki, 2012*; *Zhu, Liu & Wang, 2016*; *Liu et al., 2017*; *Miao et al., 2019*; *Ge et al., 2019*) are proposed. These schemes are the verification of single keyword search results, *Azraoui et al. (2015)* combined polynomial-based accumulators and Merkle trees to achieve conjunctive keyword verification. *Wan & Deng (2016)* used homomorphic MAC to verify the results of multi-keyword search. *Li et al. (2021)* utilized bitmap index to gain high efficiency of multi-keyword search, and verified the results by RSA accumulator. *Ge et al. (2021)* and *Liu et al. (2021)* proposed their verifiable schemes in the Internet of things. These schemes verify the results of multi-keyword search by public key cryptography primitives, which is computationally expensive and inefficient. What is more, these multi-keyword search verifiable schemes mainly focus on verifying the returned files are valid and whether the files really contains the query keywords, but they didn't ensure all files containing the query keywords are returned.

## Verifiable searchable symmetric encryption based on blockchain

In the existing SSE schemes, the verification of search results is performed by users. However, users may forge verification results for economic benefits, which damages the fairness of verification. To solve this, a flexible and feasible method is to adopt blockchain to verify search results, which uses the non-repudiable property of the blockchain to ensure the reliability and fairness of verification. *Hu et al. (2018)* built a distributed, verifiable and fair ciphertext retrieval scheme based on blockchain. *Li et al. (2019)* proposed a verifiable

**Table 1  Comparison results with existing schemes.**

| Schemes | Single-keyword | Multi-keyword | Verification | Blockchain-based |
|---|---|---|---|---|
| *Kamara, Papamanthou & Roeder, 2012* | ✓ | × | × | × |
| *Chai & Gong (2012)* | ✓ | × | ✓ | × |
| *Wang et al. (2018)* | ✓ | ✓ | ✓ | × |
| *Li et al. (2021)* | ✓ | ✓ | ✓ | × |
| *Hu et al., 2018* | ✓ | × | ✓ | ✓ |
| *Guo, Zhang & Jia, 2020* | ✓ | × | ✓ | ✓ |
| **Our scheme** | ✓ | ✓ | ✓ | ✓ |

**Figure 1  The example of bitmap.**

scheme combined blockchain and SSE, which can verify the results automatically and reduce the calculation of users. *Guo, Zhang & Jia (2020)* used the blockchain to realize the public authentication of search results, and ensures forward security of dynamic update. Although these schemes realize the fair verification of search results, but they are mainly for single keyword search, whereas there is little research on the fair verification of multi-keyword. Comparison results with existing schemes are shown in Table 1.

## PRELIMINARIES

### Bitmap

To improve search efficiency, we use the bitmap (*Spiegler & Maayan, 1985*) to build inverted index. Bitmap uses a binary string to store a set of information, which can effectively save storage space, and it has been widely used in the field of ciphertext retrieval. In our scheme, each keyword $w_i$ corresponds to a bitmap, which contains $\ell$ bits, $\ell$ is the number of files in the system, if the $i-$th document contains $w_i$ the value of $\ell$ in position $i$ is 1, otherwise 0. For example, there are four files ($f_1$, $f_2$, $f_3$, $f_4$) and two keywords ($w_1$, $w_2$), in Fig. 1, $w_1$ is contained in $f_1$ and $f_3$, $w_2$ is contained in $f_2$ and $f_3$, the bitmap of $w_1$ and $w_2$ are 1010 and 0110. If we want to search files that contains both ($w_1$ and $w_2$, we need to do AND operation on the two bitmaps, *i.e.,* $1010 \wedge 0110 = 0010$, that indicates that $f_3$ contains both $w_1$ and $w_2$.

### Blockchain

Blockchain is a distributed database, which is widely used in emerging cryptocurrencies to store transaction information such as bitcoin. The blockchain has the features of
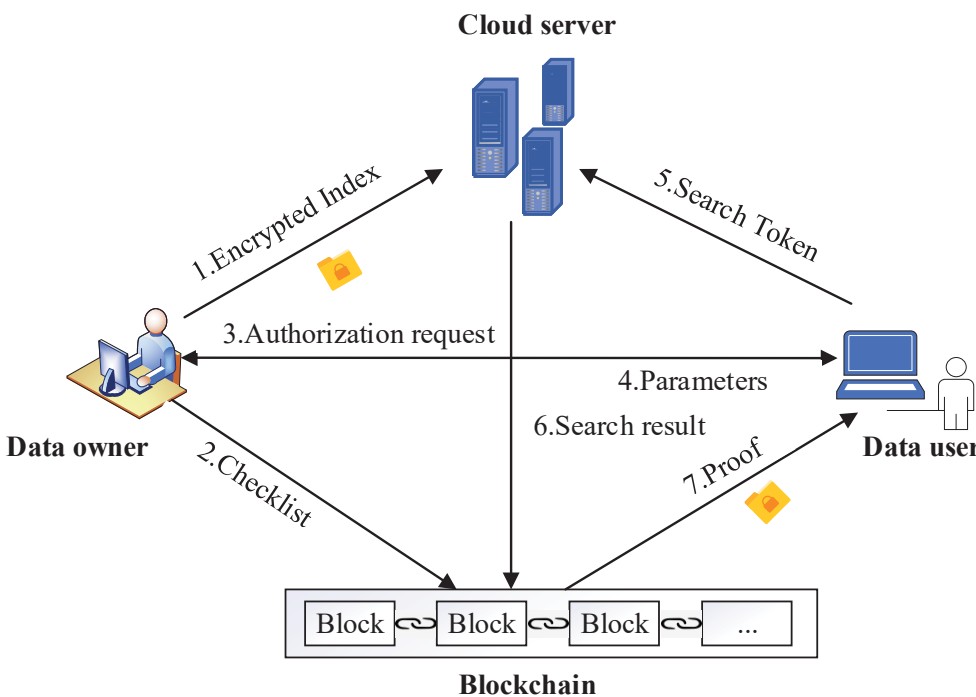

**Figure 2** System model.

decentralization, transparency and unforgeability. There is no central server in the blockchain, all nodes participate in the operation and generate the calculation results, the information stored on the blockchain can be seen by all nodes in the network. All nodes of the blockchain share the same data record, under the action of the consensus mechanism, a single node cannot modify the data stored on the chain. The above characteristics of blockchain make it suitable to be a trusted third party for fair verification.

## METHOD

### System model

The system model of our scheme is shown in Fig. 2, there are four entities in the system: data owner, cloud server, data user, blockchain. For the files **F** in the system, data owner extracts all keywords and generates a keyword set **W**. Data owner encrypts files to a database $T$, builds an encrypted index $T_\mathcal{B}$ and a checklist B, $T_\mathcal{B}$ and $T$ are sent to cloud server, $T_\mathcal{B}$ and B are sent to blockchain. When a data user joins the system, it sends an authentication request to the data owner, obtains keys and system parameters. During a query, the data user generates search token $TK_{i,Q}$ according to the keywords to be queried with the help of keys and system parameters, and then sends it to cloud server and blockchain, respectively. Cloud server provides storage services for index $T_\mathcal{B}$ and $T$. In addition, the cloud server

performs ciphertext retrieval according to the search token $TK_{i,Q}$, and sends the matched results to blockchain for verification.

To verify the search results of multiple keywords, the blockchain performs two steps: (1) benchmark. On receiving $TK_{i,Q}$, the blockchain performs multi-keyword search on the index $T_{\mathcal{B}}$ to get the identifiers *ID* of files that meets the query, then gets the corresponding hash values $\mathbb{H}$ of files from the checklist B according *ID*, and computes the benchmark *Acc* using $\mathbb{H}$; (2) verification. After receiving the results returned by cloud server, the blockchain computes the hash values $\mathbb{H}'$ of results and computes the verification value $Acc'$, then the blockchain compares *Acc* and $Acc'$ to generate the proof. The proof and search results are sent to data user, the verification is completed.

## Threat model

Like other verifiable SSE schemes (*Soleimanian & Khazaei, 2019*), we assume that the cloud server is malicious, which may return an incorrect or incomplete search result for selfish reasons, such as saving bandwidth or storage space. In addition, we assume that the data user is also untrusted, since it may forge the verification results for economic benefits. The data owner and blockchain are trusted, they execute the protocols in the system honestly.

## Algorithm definitions

Our scheme includes eight polynomial time algorithms, $\prod = \{KeyGen, Setup, ClientAuth, TokenGen, Search, Verify, UpdateToken, Update\}$, and the details are as follows:

- $K \leftarrow$ **KeyGen**$(1^{\lambda})$, takes system parameter $\lambda$ as input, and outputs system keys $K$.
- $(T, T_{\mathcal{B}}, B) \leftarrow$ **Setup**$(K, \mathbf{W}, \mathbf{F})$, takes system keys $K$, the keyword set $W$ and the set of files $F$ as input, outputs a database of encrypted files $T$, an encrypted index $T_{\mathcal{B}}$ and a checklist B.
- $(K_1, \sum) \leftarrow$ **ClientAuth**$(\mathbb{A}_i)$, takes the attribute $\mathbb{A}_i$ of user as input, outputs secret key $K_1$ and the keyword status $\sum$.
- $TK_{i,Q} \leftarrow$ **TokenGen**$(K_1, \overline{W})$, takes secret key $K_1$, a set of keywords to query $\overline{W} = \{w_1, w_2, \ldots, w_t\}$, outputs the search token $TK_{i,Q}$.
- $(R, Acc) \leftarrow$ **Search**$(T, T_{\mathcal{B}}, B, TK_{i,Q})$, takes search token $TK_{i,Q}$, the encrypted database $T$, encrypted index $T_{\mathcal{B}}$ and the checklist B as input, and outputs the search results $R$ and the benchmark *Acc*.
- $(R, proof) \leftarrow$ **Verify**$(R, Acc)$, takes the search results $R$, and the benchmark *Acc* as input, outputs the verification proof *proof* and results $R$.
- $(\tau_s, \tau_b) \leftarrow$ **UpdateToken**$(\overline{F}, W', K)$, takes the set of files to update $\overline{F}$, the set of keywords $W'$ and system keys $K = \{K_1, K_2, K_3\}$ as input, and outputs the update token $(\tau_s, \tau_b)$.
- $(T', T_{\mathcal{B}}', B') \leftarrow$ **Update**$(T, T_{\mathcal{B}}, B, \tau_s, \tau_b)$, takes encrypted database $T$, encrypted index $T_{\mathcal{B}}$ and the update token $(\tau_s, \tau_b)$ as input, outputs the updated database $T'$, updated index $T_{\mathcal{B}}'$ and the updated checklist B'.

## Security definitions

We prove the security of our scheme with the random oracle model, which can be executed by two probabilistic games $\text{Real}_{\mathcal{A}}(\lambda)$ and $\text{Ideal}_{\mathcal{A},\mathcal{S}}(\lambda)$, and we have the following definitions:

**Definition 1:** CKA2-security, for the verifiable multi-keyword search scheme $\prod = \{KeyGen, Setup, ClientAuth, TokenGen, Search, Verify, UpdateToken, Update\}$, let $\mathcal{L} = \{\mathcal{L}_{setup} \quad \mathcal{L}_{search} \quad \mathcal{L}_{update}\}$ be the leakage function, $\mathcal{A}$ is the adversary and $\mathcal{S}$ is the simulator, there are two probabilistic experiments:

$\text{Real}_{\mathcal{A}}(\lambda)$: The challenger runs $KeyGen(1^{\lambda})$ to generate secret key $K = \{K_1, K_2, K_3\}$, the adversary $\mathcal{A}$ outputs $\mathbf{F}$ and $\mathbf{W}$. The challenger triggers this experiment to run $Setup(K, \mathbf{W}, \mathbf{F})$, outputs the index $T_{\mathcal{B}}$, $T$ and $B$, which are sent to $\mathcal{A}$. $\mathcal{A}$ generates a series of adaptive queries $Q = \{q_1, q_2, \ldots, q_t\}$, for each $q_i \in Q$, the challenger generates search or update tokens, $\mathcal{A}$ receives those tokens and generates a bit $b$ as the output of this experiment.

$\text{Ideal}_{\mathcal{A}, \mathcal{S}}(\lambda)$: The adversary $\mathcal{A}$ outputs $\mathbf{F}$ and $\mathbf{W}$, the simulator $\mathcal{S}$ generates the index $T_{\mathcal{B}}$, $T$ and $B$ through $\mathcal{L}_{\text{Setup}}$, $\mathcal{A}$ receives them. $\mathcal{A}$ generates a series of adaptive queries $Q = \{q_1, q_2, \ldots, q_t\}$ , for each $q_i \in Q$, the simulator $\mathcal{S}$ generates search or update tokens with $\mathcal{L}_{\text{Search}}$ and $\mathcal{L}_{\text{Update}}$, $\mathcal{A}$ receives those tokens and generates a bit $b$ as the output of this experiment. If for any probabilistic polynomial time (PPT) adversary $\mathcal{A}$, there exist an efficient simulator $\mathcal{S}$, which satisfies that:

$$|\Pr[\text{Real}_{\mathcal{A}}(\lambda) = 1] - \Pr[\text{Ideal}_{\mathcal{A}, \mathcal{S}}(\lambda) = 1] \le negl(\lambda)$$

we say $\prod$ is $\mathcal{L}-$secure against CKA2, where *negl* is an negligible function and $\lambda$ is the security parameter.

# CONSTRUCTION

In this section, we present the construction of our scheme in detail. We take bitmap as index structure to achieve efficient search over encrypted data, and use blockchain to verify the search results. The bitmap is utilized to build the inverted index to achieve the optimal search time $\mathcal{O}(|q|)$, where $q$ is the keywords in search and $|q|$ is the number of $q$.

In our scheme, the blockchain is used to fairly verify the search results. In Setup, the data owner calculates the hash value of files, generates a checklist $B$ and saves it on the blockchain. During the verification, the blockchain smart contract computes the hash values of search results returned by the server and compares them with the existing results to obtain the verification results.

Specifically, in the single keyword setting, the blockchain stores the corresponding benchmark directly since the results corresponding to the keywords are determined. However, it's impossible in multi-keyword search because the search results are variable, which can only store the verification value of each file. To ensure the credibility of the search results, the blockchain also needs to perform multi-keyword search to obtain the search results. Therefore, we save the index $T_{\mathcal{B}}$ on the blockchain. During a query, the blockchain executes multi-keyword search to get the search results, and read the verification value $hash_i$ of each file in search results to generate the benchmark $Acc$, then the blockchain compares $Acc$ with search results returned by cloud server to complete the verification.

## Proposed construction

Our scheme contains eight algorithms $\prod = \{KeyGen, Setup, ClientAuth, TokenGen, Search,$ $Verify, UpdateToken, Update\}$, let $F : \{0,1\}^* \rightarrow \{0,1\}^m, H : \{0,1\}^* \rightarrow \{0,1\}^n$, be two Pseudo-Random Functions (PRFs), the constructions of our scheme are as follows.

$K \leftarrow$ **KeyGen**$(1^\lambda)$: This algorithm is executed by the data owner, given a security parameter $\lambda \in \mathbb{N}$, this algorithm generates the secret key $K = \{K_1, K_2, K_3\}$, where $K_1, K_2, K_3 \leftarrow \{0,1\}^\lambda$, $K_1, K_2$ are used to encrypt the bitmap index for each keyword $w_i \in \mathbf{W}$, $K_3$ is used to encrypt files $f_i \in \mathbf{F}$ and store the hash value of files.

$(T, T_\mathcal{B}, B) \leftarrow$ **Setup**$(K, W, F)$: Given a set of files $\mathbf{F}$, a set of keywords $\mathbf{W}$ and the secret keys $K$, this algorithm builds an encrypted index $T_\mathcal{B}$, a checklist $B$ and a ciphertext database $T$, as is shown in Algorithm 1. For each file $f_i \in \mathbf{F}$, $id_i$ is the identifier of $f_i$, the data owner encrypts $f_i$ by calculating $c_i \leftarrow Enc(K_3, f_i)$, and computes the hash value using $hash_i \leftarrow H(c_i)$. Then data owner stores $c_i$ and $hash_i$ in $T[l_i]$ and $B[l_i]$, respectively.

---

**Algorithm 1** Setup

---

**Require:** $K_1, K_2, K_3, W, F$

**Ensure:** $T_\mathcal{B}, B, T$

  1: **Data Owner** $DO$:

  2:    $T_\mathcal{B} \leftarrow \{\}, T \leftarrow \{\}, B \leftarrow \{\}$

  3: **for** $f_i \in F$ **do**

  4:      $l_i \leftarrow H(id_i||K_3); c_i \leftarrow Enc(K_3, f_i)$

  5:      $hash_i \leftarrow H(c_i); B[l_i] \leftarrow hash_i; T[l_i] \leftarrow c_i$

  6: **end for**

  7: **for** $w_i \in W$ **do**

  8:      Generate a bitmap index $\mathcal{B}_{w_j}$ for each $w_j$

  9:      $u_{w_i} \leftarrow F(K_1, H(w_i)); st_i \xleftarrow{\$} \{0,1\}^\lambda; t_{w_i} \leftarrow H(u_{w_i}||st_i)$

10:      $v_\mathcal{B} \leftarrow \mathcal{B}_{w_i} \oplus H(u_{w_i}||st_i); T_\mathcal{B}[t_{w_i}] \leftarrow v_\mathcal{B}; \sum[w_i] = st_i$

11: **end for**

12: send $(T_\mathcal{B}, B)$ to blockchain, send $(T, T_\mathcal{B})$ to cloud server

---

For each keyword $w_i \in \mathbf{W}$, data owner generates a bitmap $\mathcal{B}_{w_i}$, if $id_j$ contains keyword $w_i$, then $\mathcal{B}_{w_i}[m] = 1$, where $m = H(id_j||K_3)$, and the other positions of $\mathcal{B}_{w_i}$ are all $0$'s. The data owner encrypts $\mathcal{B}_{w_i}$ through $v_\mathcal{B} \leftarrow \mathcal{B}_{w_i} \oplus H(t_w||st_{i+1})$, and store $v_\mathcal{B}$ in $T_\mathcal{B}[t_w]$. At the end of the Setup, $(T_\mathcal{B}, B)$ and $((T), T_\mathcal{B})$ are sent and stored on blockchain and cloud server, respectively.

$(K_1, \sum) \leftarrow ClientAuth(\mathbb{A}_i)$: It needs to register to the data owner when a new data user who wants to query files on the cloud server joins the system. The data user submits attribute $\mathbb{A}_i$ to the data owner through this algorithm to obtain the keyword status $\sum$ and the key $K_1$.

$TK_{i,Q} \leftarrow$ **TokenGen**$(K_1, \overline{W})$: It takes the key $K_1$ and the set of keywords to query $\overline{W} = \{w_1, w_2, \ldots, w_t\}$ as input, output a search token $TK_{i,Q}$, as is shown in Algorithm 2. For each keyword $w_i \in \overline{W}$, the data user computes the position $l_{w_i}$ of $w_i$ in index $T_\mathcal{B}$ as

$l_{w_i} \leftarrow H(u_{w_i}||st_i)$, where $u_{w_i} \leftarrow F(K_1, H_1(w_i))$, $st_i \leftarrow \sum[w_i]$. Data user sends $TK_{i,Q}$ to cloud server and blockchain, respectively.

$(R, Acc) \leftarrow$ **Search**$(T, T_{\mathcal{B}}, B, TK_{i,Q})$: This algorithm takes search token $TK_{i,Q}$, index $T_{\mathcal{B}}$ and ciphertext database $T$ as input, and outputs search results $R$. On receiving the search token, the cloud server and blockchain perform the same operations for multi-keyword search. They all parse out the position $l_{w_i}$ of the keyword in the token $TK_{i,Q}$, and get the bitmap $\mathcal{B}_{w_i}$ through $\mathcal{B}_{w_i} \leftarrow v_{\mathcal{B}} \oplus H(K_{w_i}||l_i)$, $v_{\mathcal{B}} \leftarrow T_{\mathcal{B}}[l_{w_i}]$. To achieve multi-keyword search, they compute $\mathcal{B} = \mathcal{B}_1 \wedge \mathcal{B}_2 \wedge \ldots \wedge \mathcal{B}_t$, the cloud server gets files in $T$ according to $\mathcal{B}$ with regard to $\mathcal{B}[i] = 1$, and sends them to the blockchain to verify. Similarly, the blockchain gets hash values $\{hash_1, hash_2, \ldots, hash_s\}$ of files in $B$ according to $\mathcal{B}$, computes $Acc = hash_1 \oplus hash_2 \oplus \cdots \oplus hash_s$ as the benchmark for verification, and the details are shown in Algorithm 2.

---

**Algorithm 2** Search

---

**Require:** $K_1, \overline{W} = \{w_1, w_2, \ldots, w_t\}$, T, $T_{\mathcal{B}}$, B
**Ensure:** $TK_{i,Q}, R, Acc$

1: **Data user:**
2: **for** $w_i \in \overline{W}$ **do**
3:     $st_i \leftarrow \sum[w_i]$; $u_{w_i} \leftarrow F(K_1, H(w_i))$; $l_{w_i} \leftarrow H(u_{w_i}||st_i)$
4: **end for**
5: **return** $TK_{i,Q} \leftarrow (l_{w_1}, l_{w_2}, \ldots, l_{w_t})$
6: Send $TK_{i,Q}$ to cloud server and blockchain
7: **Server, Blockchain:**
8: **for** $l_{w_i} \in TK_{i,Q}$ **do**
9:     $v_{\mathcal{B}} \leftarrow T_{\mathcal{B}}[l_{w_i}]$; $\mathcal{B}_{w_i} \leftarrow v_{\mathcal{B}} \oplus H(l_{w_i})$
10: **end for**
11: $\mathcal{B} = \mathcal{B}_1 \wedge \mathcal{B}_2 \wedge \ldots \wedge \mathcal{B}_t$
12: **Server:**
13: gets ciphertext $R = \{c_1, c_2, \ldots, c_s\}$ form T
14: **Blockchain:**
15: gets checklist $L = \{hash_1, hash_2, \ldots, hash_s\}$ from B with $\mathcal{B}$
16: $Acc = hash_1 \oplus hash_2 \oplus \cdots \oplus hash_s$

---

$(R, proof) \leftarrow$ **Verify**$(R, Acc)$: This algorithm takes search results $R$ and benchmark $Acc$ as input, outputs search results $R$ and $proof$, and the verify process is shown in Algorithm 3. To verify the integrity of files, the data owner calculates the hash value of each file through $hash_i \leftarrow H(c_i)$ in the Setup, and adds $hash_i$ to the checklist $B$, then $B$ is sent to the blockchain. Through algorithm *Search*, the blockchain gets the search result of multiple keywords, obtains the hash value of each file in the result from $B$, and computes the benchmark $Acc$. To verify the search results, the blockchain calculates $H_{\overline{W}}$ of $R$ and compares it with $Acc$.

In Algorithm 3, for all ciphertexts $c_i \in R$, blockchain computes $H_{\overline{W}} \leftarrow H_{\overline{W}} \oplus H(c_i)$, where $H(c_i)$ denotes the hash value of $c_i$. Blockchain compares $H_{\overline{W}}$ and $Acc$, if they are

---

**Algorithm 3** Verify

**Require:** $R, Acc$

**Ensure:** proof, Result

1: **Blockchain:**
2: $H_{\overline{W}} \leftarrow \phi$
3: **for** $c_i \in R$ **do**
4:      $H_{\overline{W}} \leftarrow H_{\overline{W}} \oplus H(c_i)$
5: **end for**
6: **if** $H_{\overline{W}} = Acc$ **then**
7:      proof = true, Result $\leftarrow R$
8: **else**
9:      proof = false, Result $\leftarrow \phi$
10: **end if**
11: sends (proof, Result) to data user

---

equal, the proof is true, otherwise false. At last, the search results $R$ and proof are sent to data user. During the verification, $Acc$ is calculated through the hash value stored on the blockchain, due to the unforgeability of blockchain, thus $Acc$ is unforgeable. In addition, the verification is completed by the blockchain, so the proof is also unforgeable, which ensures the fairness of verification.

$(\tau_s, \tau_b) \leftarrow$ **UpdateToken**$(\overline{F}, W', K)$: The data owner generates an update token through this algorithm, which takes files $\overline{F}$, a keyword set $W'$ and secret key $K$ as input, and outputs update token$(\tau_s, \tau_b)$. For files $f_k \in \overline{F}$, the data owner encrypts and calculates the hash value of $f_k$ by $c_k \leftarrow \text{Enc}(K_3, f_k)$ and $hash_k \leftarrow H(c_k)$, respectively. For keywords $W' = \{w_1, w_2, \ldots, w_s\}$ that contained in $f_k$, the data owner generates a bitmap $\mathcal{B}_{w_j}$ for each $w_j \in W'$, and encrypts $\mathcal{B}_{w_j}$ with $v_{\mathcal{B}} \leftarrow \mathcal{B}_{w_j} \oplus H(l_{w_j} || st)$, where $l_{w_j} \leftarrow H(u_{w_j} || st)$, $u_{w_j} \leftarrow F(K_1, H(w_j))$, $st \leftarrow F(K_2, st_0)$.

$(T', T_{\mathcal{B}}', B') \leftarrow$ **Update**$(T, T_{\mathcal{B}}, B, \tau_s, \tau_b)$: This algorithm takes encrypted database $T$, index $T_{\mathcal{B}}$, checklist $B$, update token $(\tau_s, \tau_b)$ as input, and outputs updated database $T'$, updated index $T_{\mathcal{B}}'$ and updated checklist $B'$. The details are shown in Algorithm 4.

## Forward security

As described above, dynamic update is the foundation function of an SSE scheme, and forward security is an indispensable component of dynamic update. In Algorithm 4, when updating a file $f_i$ that contains keyword $w_j$, the data owner retrieves the previous state $st_0$ from the local state store $\sum$, and generates a new state $st$ through $st \leftarrow F(K_2, st_0)$, where $F$ is a pseudo random function and $K_2$ is kept in local. To search a keyword $w_j$, the data user retrieves the current state $st_0$ from $\sum$, with $st_0$ data user generates a token to be sent to the cloud server and blockchain. Without the key $K_2$, the server cannot compute the current state $st$ from a previous state $st_0$, therefore it cannot get the current token from a previous, considering that the newly added file $f_i$ corresponds to the current token, that means the previous tokens cannot match $f_i$, then forward security is achieved.

---

**Algorithm 4** Update

**Require:** $\overline{F}$, $K = \{K_1, K_2, K_3\}$, W′, T, T$_\mathcal{B}$, B
**Ensure:** $\tau_s, \tau_b,$ T′, T$_\mathcal{B}$′, B′

1: **Data owner:**
2: **for** $f_k \in \overline{F}$ **do**
3:   $l_k \leftarrow H(id_k || K_3)$, $c_k \leftarrow \text{Enc}(K_3, f_k)$, $hash_k \leftarrow H(c_k)$
4:   $f_k$, W′ $= \{w_1, w_2, \ldots, w_s\}$
5:   **for** $w_j \in$ W′ **do**
6:    generates a bitmap index $\mathcal{B}_{w_j}$ for $w_j$
7:    **if** $\sum[w_j] = \phi$ **then**, then $st_0 \xleftarrow{\$} \{0,1\}^\lambda$
8:    **else**
9:     $st_0 \leftarrow \sum[w_j]$, $st \leftarrow F(K_2, st_0)$
10:     $u_{w_j} \leftarrow F(K_1, H(\text{w}_j))$, $l_{w_j} \leftarrow H(u_{w_j} || st)$
11:     $v_{\mathcal{B}_j} \leftarrow \mathcal{B}_{w_j} \oplus H(u_{w_j} || st)$, $\sum[w_j] = st$
12:    **end if**
13:   **end for**
14:   return $\tau_s = \{(l_k, c_k), (l_{w_j}, v_{\mathcal{B}_j})\}$, $\tau_b = \{(l_k, hash_k), (l_{w_j}, v_{\mathcal{B}_j})\}$
15: **end for**
16: **Server:** $T[l_k] \leftarrow c_k$, $T_\mathcal{B}[l_{w_j}] \leftarrow v_{\mathcal{B}_j}$, $T' \leftarrow T$, $T_\mathcal{B}' \leftarrow T_\mathcal{B}$
17: **Blockchain:** $B[l_k] \leftarrow hash_k$, $T_\mathcal{B}[l_{w_j}] \leftarrow v_{\mathcal{B}_j}$, $B' \leftarrow B$, $T_\mathcal{B}' \leftarrow T_\mathcal{B}$

---

## SECURITY ANALYSIS

In this section, we analysis the security of our scheme. For the scheme $\prod = KeyGen, Setup,$ $ClientAuth, TokenGen, Search, Verify, UpdateToken, Update$ with the leakage function $\mathcal{L} = \{\mathcal{L}_{\text{setup}}, \mathcal{L}_{\text{search}}, \mathcal{L}_{\text{update}}\}$, we prove that our scheme is $\mathcal{L}-$ secure against CKA2 by proving that $\text{Real}_\mathcal{A}(\lambda)$ and $\text{Ideal}_{\mathcal{A}, \mathcal{S}}(\lambda)$ are computationally indistinguishable.

**Theorem 1.** Our scheme $\prod$ is $\mathcal{L}-$ secure against CKA2, if the encryption algorithm is secure against chosen-plaintext attacks and the pseudo-random function $F$ and $H$ are secure pseudo-random.

**Proof:** We use a probabilistic polynomial time simulator $\mathcal{S}$ to simulate indexes and a series of tokens. For a PPT adversary $\mathcal{A}$, we prove Theorem 1 by the computational indistinguishability between $\text{Real}_\mathcal{A}(\lambda)$ and $\text{Ideal}_{\mathcal{A}, \mathcal{S}}(\lambda)$. In $\text{Real}_\mathcal{A}(\lambda)$, $\mathcal{A}$ gets indexes ($T_\mathcal{B}$, $T$ and $B$), searches token $TK_{i,Q}$ and updates token ($\tau_s$, $\tau_b$) by running Setup, TokenGen and UpdateToken; in $\text{Ideal}_{\mathcal{A}, \mathcal{S}}(\lambda)$, $\mathcal{A}$ gets indexes ($T_\mathcal{B}'$, $T'$ and $B'$), searches token $TK_{i,Q}'$ and updates token ($\tau_s'$, $\tau_b'$) by running $\mathcal{L}_{Setup}, \mathcal{L}_{Search}, \mathcal{L}_{Update}$. We prove that $\text{Real}_\mathcal{A}(\lambda)$ and $\text{Ideal}_{\mathcal{A}, \mathcal{S}}(\lambda)$ are computational indistinguishable by proving that ($T_\mathcal{B}$, $T$, $B$, $TK_{i,Q}$, $\tau_s$, $\tau_b$) and ($T_\mathcal{B}'$, $T'$, $B'$, $TK_{i,Q}'$, $\tau_s'$, $\tau_b'$) are indistinguishable.

**Simulating index.** $\mathcal{S}$ initializes three empty tables: $T'$, $B'$, $T_\mathcal{B}'$, which are used to store file ciphertexts, verification values and bitmaps, respectively. $\mathcal{S}$ randomly selects a string $f_i'$ of length $|f_i|$, and encrypts it through $c_i' \leftarrow \text{Enc}(K_3, f_i')$, where $K_3$ is randomly sampled from $\{0,1\}^\lambda$. $\mathcal{S}$ maintains three mappings: H, U and L, H stores $(id_i || K_3, \ell_i')$, U stores

$(H(w_i), u_{w_i}')$, and the mapping L stores $(u_{w_i}'||st_i, t_{w_i}')$. H, U and L are used and updated by the generation of search and update token. $\mathcal{S}$ computes the hash value $hash_i' \leftarrow H(c_i')$, $c_i'$ is stored in $T'[l_i']$ and $hash_i'$ is stored in $B'[l_i']$. $\mathcal{S}$ selects a string $v_\mathcal{B}'$ of length $|v_\mathcal{B}|$, and stores it in $T_\mathcal{B}'[t_{w_i}] \leftarrow v_\mathcal{B}$.

$T'$, $B'$ and $T_\mathcal{B}'$ are simulated by $\mathcal{S}$ through the leakage $\mathcal{L}_{Setup}$, the difference between $(T_\mathcal{B}', T', B')$ and $(T_\mathcal{B}, T, B)$ is the generation of $(f_i', c_i', v_\mathcal{B}')$. In ideal environment, $(f_i', c_i', v_\mathcal{B}')$ are randomly selected, since our encryption algorithm is secure against CKA2, $F$ and $H$ are secure pseudo-random functions, therefore , the probability that the adversary $\mathcal{A}$ can distinguish between the real environment and the ideal environment is negligible.

**Simulating search token.** For the keyword $w_i$ to query, $\mathcal{S}$ gets $u_{w_i}'$ from the mapping U through calculating $H(w_i)$, $\mathcal{S}$ checks whether $u_{w_i}'$ is contained in U, if so returns the corresponding entity, otherwise randomly picks a $u_{w_i}'$ in $\{0,1\}^\ell$ and stores $(H(w_i), u_{w_i}')$ in U. Similarly, the experiment gets $l_{w_i}'$ from L by $L[u_{w_i}'||st_i]$, the search token $TK_{i,Q}' = l_{w_i}'$. Under the assumption that $F$ and $H$ are secure pseudo-random functions, the adversary $\mathcal{A}$ cannot distinguish $TK_{i,Q}$ and $TK_{i,Q}'$.

**Simulating update token.** For file $f_k$ to be added, $\mathcal{S}$ first randomly selects a bit string $c_k'$ of length $|f_k|$, and encrypts it through $c_k' \leftarrow Enc(K_3, f_k')$. $\mathcal{S}$ computes the hash value $hash_k' \leftarrow H(c_k')$, $c_k'$ is stored in $T'[l_k']$ and $hash_k'$ is stored in $B'[l_k']$, where $l_k'$ is obtained from the mapping H. $\mathcal{S}$ maintains a mapping E, which stores $(st_0, st)$, if there is no corresponding entity for $st$, it randomly picks a $st$ in $\{0,1\}^l$, otherwise it returns the corresponding entity. $\mathcal{S}$ gets $u_{w_i}'$ and $l_{w_i}'$ as in search token, selects a string $v_{\mathcal{B}_j}'$ of length $|v_{\mathcal{B}_j}|$, and stores it in $T_\mathcal{B}'[l_{w_j}'] \leftarrow v_{\mathcal{B}_j}'$. The update token $(\tau_s' = \{(l_k', c_k'), (l_{w_j}', v_{\mathcal{B}_j}')\}$, $\tau_b' = \{(l_k', hash_k'), (l_{w_j}', v_{\mathcal{B}_j}')\})$ and $(\tau_s = \{(l_k, c_k), (l_{w_j}, v_{\mathcal{B}_j})\}$, $\tau_b = \{(l_k, hash_k), (l_{w_j}, v_{\mathcal{B}_j})\})$ are indistinguishable for the adversary $\mathcal{A}$.

In such a way, $(T_\mathcal{B}, T, B, TK_{i,Q}, \tau_s, \tau_b)$ and $(T_\mathcal{B}', T', B', TK_{i,Q}', \tau_s', \tau_b')$ are indistinguishable for $\mathcal{A}$, and it means for a PPT adversary $\mathcal{A}$, the probability of distinguishing between $Real_\mathcal{A}(\lambda)$ and $Ideal_{\mathcal{A},\mathcal{S}}(\lambda)$ is negligible, so we have:

$$|\Pr[Real_\mathcal{A}(\lambda) = 1] - \Pr[Ideal_{\mathcal{A},\mathcal{S}}(\lambda) = 1]| \leq negl(\lambda)$$

Therefore, our scheme satisfies CKA2-security.

## PERFORMANCE EVALUATION

In this section, we evaluate the performance of our scheme by constructing a series of experiments, and compared the experimental results with *Li et al. (2021)* and *Guo, Zhang & Jia (2020)*. Since *Guo, Zhang & Jia (2020)* do not support multi-keyword search over encrypted data, we compared our scheme with (*Li et al., 2021*) which supports multi-keyword search. We also compared our scheme with (*Guo, Zhang & Jia, 2020*) in terms of dynamic operations.

We deploy our experiments on a local machine with an Intel Core i7-8550U CPU of 1.80 GHz, 8GB RAM. We use HMAC-SHA-256 for the pseudo-random functions, SHA-256 for the hash function. We use AES as the encryption algorithm to encrypt files.

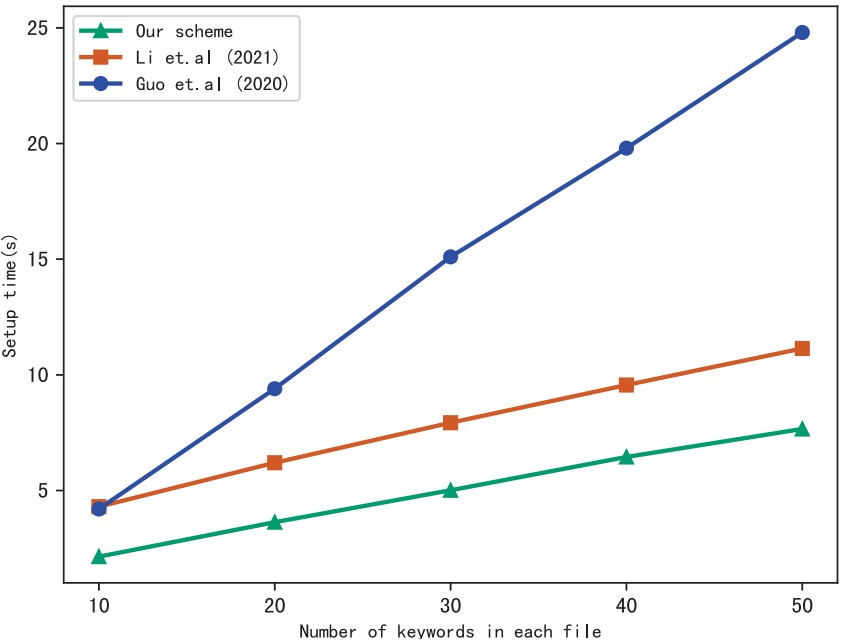

**Figure 3** Performance of the setup, files = 3,137.

We implement the algorithms in data owner, data user and server using Python and construct the smart contract using Solidity, and the smart contract is tested in with the Ethereum blockchain using a local simulated network TestRPC.

For the dataset, we adopt a real-world dataset, Enron email dataset (*WC., 2015*), which contains more than 517 thousand documents. We utilize the Porter Stemmer to extract more than 1.67 million keywords and filter that meaningless keywords, such as of, the. At last, we build an inverted index with those keywords to improve the search efficiency of the experiment.

## Evaluation of setup

In setup phase, data owner encrypts the files, calculates the initial verification values of ciphertexts, generates the bitmap indexes of keywords, stores them in T, B and $T_{\mathcal{B}}$, respectively.

First, we compare the setup time of our scheme with *Li et al. (2021)* and *Guo, Zhang & Jia (2020)*, the setup time is related to the number of files in the index and the number of keywords included in each file. Figure 3 shows the setup time with different number of keywords in each file while the number of files is fixed at 3137, Fig. 4 shows the setup time with different number of files when the number of keywords in each file is fixed at 20. Both figures show that the setup time is affected by the number of keywords in each file and the number of files, and the setup time increases linearly concerning the number of keywords and files.

Furthermore, Figs. 3 and 4 illustrate that our scheme is more efficient than *Li et al. (2021)* and *Guo, Zhang & Jia (2020)* under the same condition in setup time. Since *Guo,*

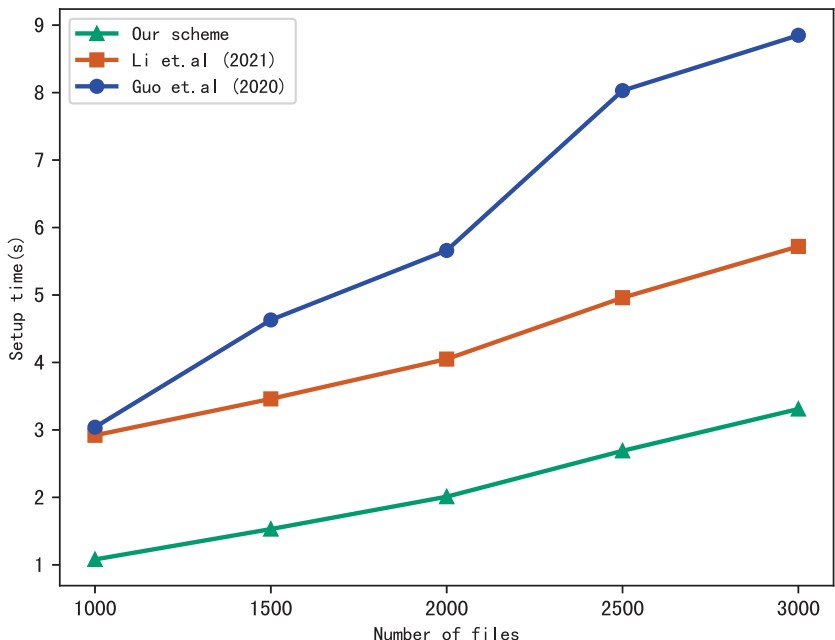

**Figure 4  Performance of the setup, keywords = 20.**

*Zhang & Jia (2020)* utilizes the linked list instead of bitmap to build the index, it requires more time than the other schemes. Our scheme takes less time than *Li et al. (2021)*, the reason is that *Li et al. (2021)* adopts RSA accumulator based on public key encryption to verify multi-keyword search results, in contrast, our scheme utilizes hash functions to verify search results, which reduces the computational overhead greatly.

## Evaluation of search

For the performance of our scheme, we compare the search time of our scheme with *Li et al. (2021)*. Moreover, to better evaluate the performance of the scheme in multi-keyword search, we perform two settings in a query: 5 keywords and 10 keywords, respectively. In figures, the suffix of the icon indicates the number of keywords in a query, *i.e.,* our scheme_5 indicates the search time spent in our scheme during a query which contains five keywords: our scheme_10 indicates the search time spent in our scheme during a query which contains 10 keywords, similarly, *Li et al. (2021)*_5 and *Li et al. (2021)*_10 indicates the search time spent in *Li et al. (2021)* during a query which contains five keywords and 10 keywords, respectively.

Figure 5 shows the search time with different number of keywords in each file when the number of files is fixed at 3,137, and Fig. 6 shows the search time with different number of files when the number of keywords in each file is fixed at 20. Both figures show that the search time is affected by the number of keywords in each file and the number of files, and the search time increases sub-linearly with the number of keywords and files.

From Figs. 5 and 6, we can see that the more keywords included in a query, the more time it takes, this is because the more keywords, the search algorithm spends more time to

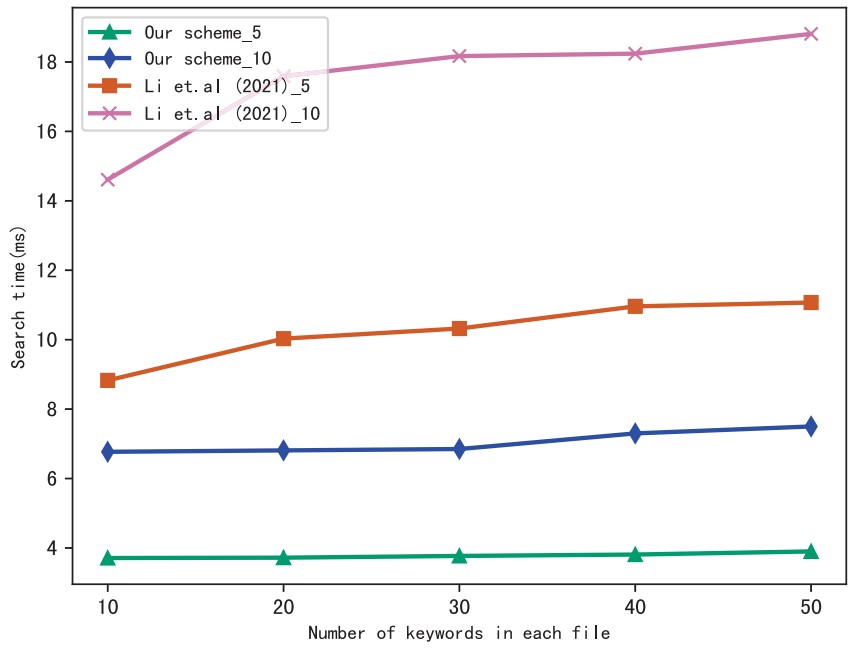

**Figure 5** **Performance of the search, files = 3,137.**

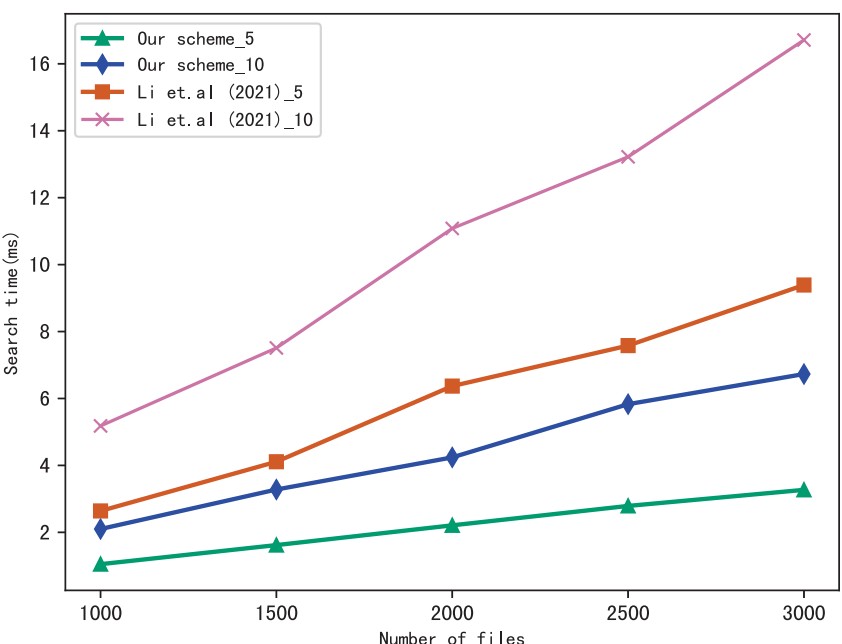

**Figure 6** **Performance of the search, keywords = 20.**

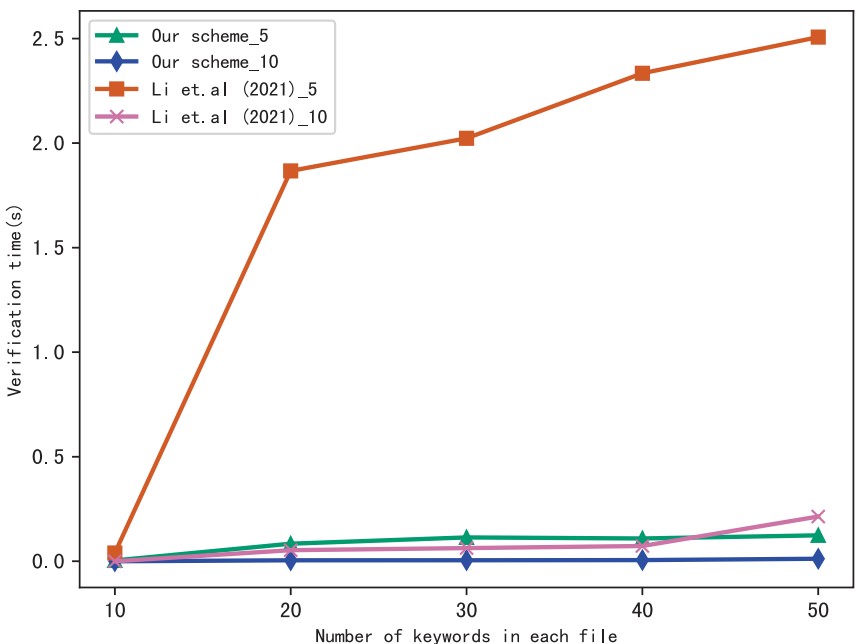

**Figure 7** Performance of the verification, files = 3,137.

calculate matched files. Another conclusion can be drawn that our scheme is more efficient than *Li et al. (2021)* in search, the reason is that the same as the setup algorithm, *Li et al. (2021)* takes more time to calculate the verification values.

## Evaluation of verify

Here, we evaluate the performance of our scheme in verification, we verify the results of searching for 5 keywords and 10 keywords respectively, and compares the verification time with *Li et al. (2021)*, the comparison results are shown in Figs. 7 and 8. Figure 7 shows the verification time with different number of keywords in each file when the number of files is fixed at 3,137, and Fig. 8 shows the verification time with different number of files when the number of keywords in each file is fixed at 20. From those two figures, we can see that the verification time is affected by the number of keywords in each file and the number of files, the verification time increases with the number of keyword and files.

Both figures shows that our scheme gains a higher verification efficiency than *Li et al. (2021)*, the reason is that *Li et al. (2021)* takes additional time to compute $\mathcal{B}_{f_i} = y_i \oplus u_i$, where $u_i = F(K_{f_i}||r_i)$, $K_{f_i} = G(K_3, f_i)$. In addition, the initial verification values in *Li et al. (2021)* are stored in untrusted server and the verification is performed by the data user, both the server and the user may forge the verification results, while in our scheme, the values are stored in blockchain and the verification is performed by blockchain, cannot be tampered with, hence, our scheme is more fair and secure in verification.

Dynamic update is the important function in SSE, so we evaluate the performance of our scheme in dynamic update by adding a file containing multiple keywords. Figures 9 and 10 show the performance of our scheme, *Li et al. (2021)* and *Guo, Zhang & Jia (2020)*

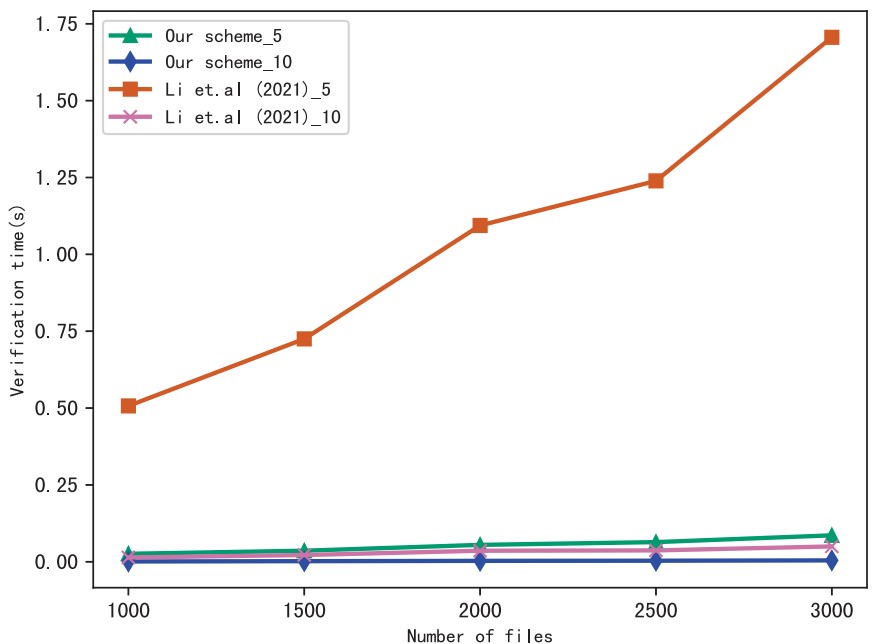

**Figure 8** **Performance of the verification, keywords = 20.**

in update time, _5 and _10 indicate that the update document contains 5 keywords and 10 keywords, respectively. We observe that the update time increases with the number of files, since the more files, the longer of the bitmap corresponding to a keyword, then the update algorithm performs more operations when calculating $v_{\mathcal{B}} \leftarrow \mathcal{B}_{w_j} \oplus H(u_{w_j}||st)$. Moreover, the update time is related to the number of keywords contained in the update file, since the more keywords the file contains, the more indexes to update.

## CONCLUSIONS

In this paper, we present an efficient verifiable multi-keyword search SSE scheme based on blockchain, which accomplishes efficient multi-keyword search and verification. In our scheme, the yardstick of the file is stored on the blockchain, and the verification of the search results is also completed by the blockchain, thus the fairness and reliability of the verification can be ensured. In addition, our solution supports the dynamic update of files and guarantees forward security during the update. Formal security analysis and experimental results show that our scheme is CKA2-security and efficient. Our scheme can be widely used in cloud storage systems such as data outsourcing, cloud-based IoT (*Ge et al., 2021*), medical cloud data (*Li et al., 2017*), etc., helping to achieve efficient multi-keyword searches, and ensuring the integrity and credibility of search results.

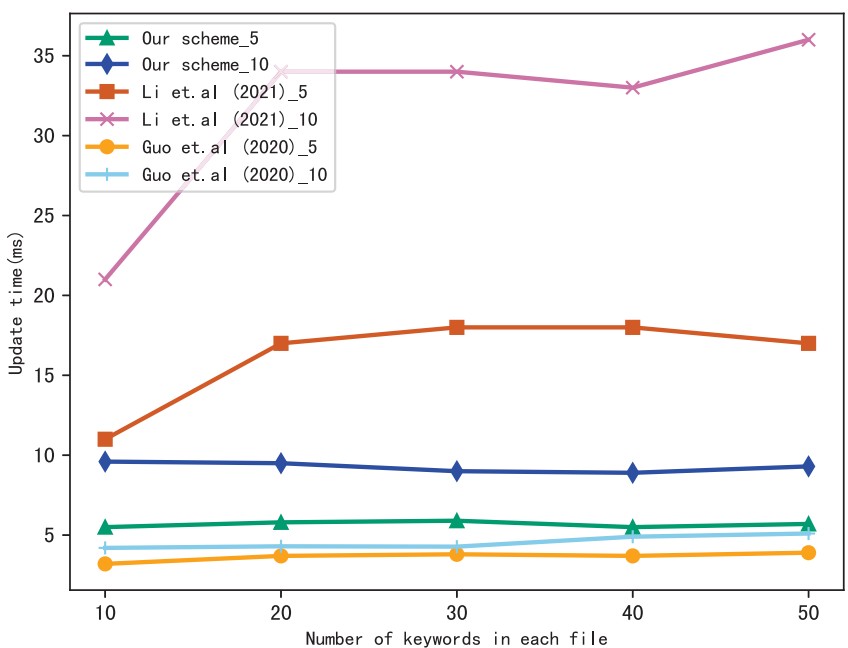

**Figure 9   Performance of the update, files = 3,137.**

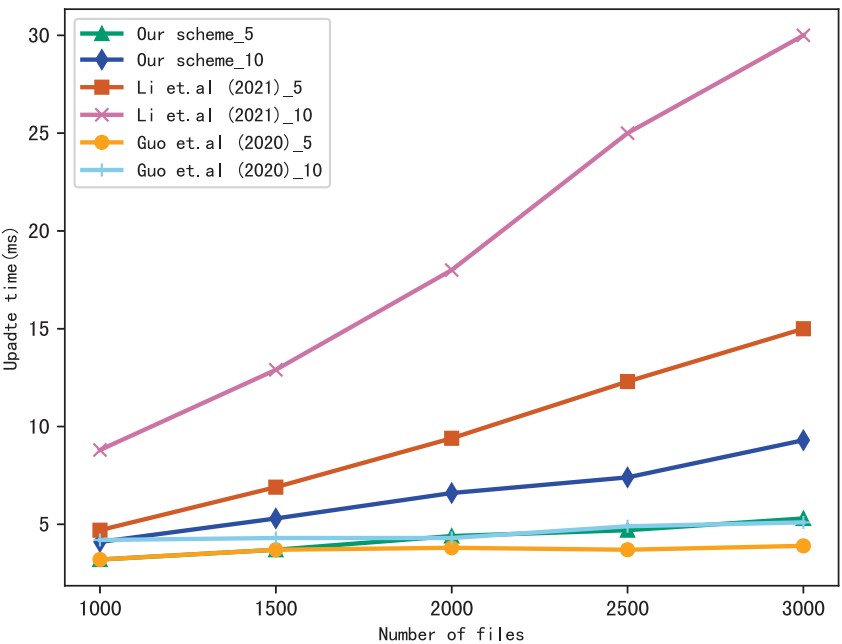

**Figure 10   Performance of the update, keywords = 20.**

### Funding

This work was supported by the Natural Science Foundation of Beijing Municipality under Grant M21039. The funders had no role in study design, data collection and analysis, decision to publish, or preparation of the manuscript.

### Grant Disclosures

The following grant information was disclosed by the authors:
The Natural Science Foundation of Beijing Municipality: M21039.

### Competing Interests

The authors declare there are no competing interests.

### Author Contributions

- Wanshan Xu conceived and designed the experiments, performed the experiments, analyzed the data, performed the computation work, prepared figures and/or tables, and approved the final draft.
- Jianbiao Zhang conceived and designed the experiments, authored or reviewed drafts of the paper, and approved the final draft.
- Yilin Yuan conceived and designed the experiments, analyzed the data, prepared figures and/or tables, and approved the final draft.
- Xiao Wang analyzed the data, prepared figures and/or tables, authored or reviewed drafts of the paper, and approved the final draft.
- Yanhui Liu performed the experiments, performed the computation work, authored or reviewed drafts of the paper, and approved the final draft.
- Muhammad Irfan Khalid conceived and designed the experiments, authored or reviewed drafts of the paper, and approved the final draft.

### Data Availability

The code is available in the Supplementary File.

### Supplemental Information

Supplemental information for this article can be found online at http://dx.doi.org/10.7717/peerj-cs.930#supplemental-information.

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
