# Peer review of "Towards efficient verifiable multi-keyword search over encrypted data based on blockchain"

_PeerJ Computer Science, doi:10.7717/peerj-cs.930_

## Round 0.1 · original submission · Major Revisions

The presentation and experimental section in the paper must be improved.

Reviewer 1 ·

Basic reporting

The authors please make any changes to improve the readability of the paper.
The reference is insufficient. The authors need to add popular related articles to support the proposed arguments.
The authors need to add more detailed contents to enrich the proofing process.

Experimental design

No comments.

Validity of the findings

The paper proposed a conceptual approach. If possible, the authors please verify the applicability of the proposed algorithm.

Reviewer 2 ·

Basic reporting

no comment

Experimental design

no comment

Validity of the findings

no comment

Annotated reviews are not available for download in order to protect the identity of reviewers who chose to remain anonymous.

Reviewer 3 ·

Basic reporting

In this article, the author proposes a blockchain-based multi-keyword verifiable symmetric searchable encryption scheme. Compared with previous work, it improves the search efficiency, ensures the fairness of the search, and can perform multi-key Word search result verification. Finally, the author uses experiments to prove that this method is safe and effective in practical applications. Here are some comments for the authors to improve the quality of this manuscript.

Experimental design

The experimental design should be more detailed, and the pictures will be more convincing

Validity of the findings

This is the same problem, adding your pictures will make your findings more convincing

Additional comments

1. I don’t know if it’s due to the system or other reasons. I didn’t see your picture in the article.
2. In the introduction, I think you can write more compactly. For example, line 45, the shortcomings of SSE are introduced before, and dynamic updates are also very important afterwards.
3. It is recommended to use subtitles to make your article clearer.
4. Please use correct template of this journal.

---

## Round 0.2 · Minor Revisions

In the final version, please address the remaining minor concerns of the reviewers.

Reviewer 1 ·

Basic reporting

The revised draft made progress.

Experimental design

It is acceptable.

Validity of the findings

It is acceptable.

Reviewer 3 ·

Basic reporting

The authors modify the manuscript according to the comments. It improves a lot compared to the previous version. However, there are still some details to be corrected to make this manusript better. Would the authors use the latex template if possible since the typewritting is not so neat, especially for the equations.

Experimental design

The proof of the scheme is sound. The simulation results can be improved if possible.

Validity of the findings

The novelty is fine. Are there any data in the simulation? Please list them or clain the origin if possible.

---

## Round 0.3 · accepted · Accept

No further change is requested.

Reviewer 2 ·

Basic reporting

The revised draft made progress.

Experimental design

It is acceptable.

Validity of the findings

It is acceptable.

Reviewer 3 ·

Basic reporting

I recommend to accept it in the current version.

Experimental design

The experiments in this paper are fine.

Validity of the findings

The findings are sound.